# miR-590-3p Targets Cyclin G2 and FOXO3 to Promote Ovarian Cancer Cell Proliferation, Invasion, and Spheroid Formation

**DOI:** 10.3390/ijms20081810

**Published:** 2019-04-12

**Authors:** Mohamed Salem, Yanan Shan, Stefanie Bernaudo, Chun Peng

**Affiliations:** 1Department of Biology, York University, Toronto, ON M3J 1P3, Canada; msalem@yorku.ca (M.S.); shan1104@yorku.ca (Y.S.); sbr@yorku.ca (S.B.); 2Centre for Research on Molecular Interactions, York University, Toronto, ON M3J 1P3, Canada

**Keywords:** miR-590-3p, ovarian cancer, spheroid formation, Cyclin G2, FOXO3, β-catenin

## Abstract

Ovarian cancer is the leading cause of death from gynecological cancers. MicroRNAs (miRNAs) are small, non-coding RNAs that interact with the 3′ untranslated region (3′ UTR) of target genes to repress their expression. We have previously reported that miR-590-3p promoted ovarian cancer growth and metastasis, in part by targeting Forkhead box A (FOXA2). In this study, we further investigated the mechanisms by which miR-590-3p promotes ovarian cancer development. Using luciferase reporter assays, real-time PCR, and Western blot analyses, we demonstrated that miR-590-3p targets cyclin G2 (CCNG2) and Forkhead box class O3 (FOXO3) at their 3′ UTRs. Silencing of CCNG2 or FOXO3 mimicked, while the overexpression of CCNG2 or FOXO3 reversed, the stimulatory effect of miR-590-3p on cell proliferation and invasion. In hanging drop cultures, the overexpression of mir-590 or the transient transfection of miR-590-3p mimics induced the formation of compact spheroids. Transfection of the CCNG2 or FOXO3 plasmid into the mir-590 cells resulted in the partial disruption of the compact spheroid formation. Since we have shown that CCNG2 suppressed β-catenin signaling, we investigated if miR-590-3p regulated β-catenin activity. In the TOPFlash luciferase reporter assays, mir-590 increased β-catenin/TCF transcriptional activity and the nuclear accumulation of β-catenin. Silencing of β-catenin attenuated the effect of mir-590 on the compact spheroid formation. Taken together, these results suggest that miR-590-3p promotes ovarian cancer development, in part by directly targeting CCNG2 and FOXO3.

## 1. Introduction

Epithelial ovarian cancer (EOC) is the most lethal cancer of the female reproductive tract and is one of the leading causes of cancer death in women [1,2]. The majority of EOC cases are not detected until the cancer cells have spread to other abdominal or distal organs [1,2]. A major route of EOC metastasis is through the formation of ascites [3]. Within the ascites fluid, malignant cells may present as single cells or aggregated spheroids. It is believed that the change in cell behavior required for compact spheroid formation may contribute to the promotion of ovarian cancer progression [4] and resistance to chemotherapies [5]. To study this mechanism *in vitro*, researchers have developed the hanging drop culture system to monitor spheroid formation [6]. It has been reported that the formation of tighter and more compact spheroids is associated with more aggressive EOC phenotypes [7].

MicroRNAs (miRNAs) are small, non-coding RNAs that play important roles in regulating gene expression [8,9]. miRNAs are typically 18–22 nucleotides in length and selectively target messenger RNA (mRNA) through complementary binding to the 3′ UTR of target mRNAs. The complementarity between miRNA and its target mRNA sequence defines how the target mRNA is processed. miRNA with the help of an AGO2-dependent slicer, which is a component of the RNA-induced silencing complex (RISC), can cleave highly complementary target mRNAs, whereas miRNAs with lower complementarity to their target mRNAs reduce mRNA stability and inhibit translation [8]. miR-590-3p is generated from its precursor, mir-590, which is processed from the intron of the eukaryotic translation initiation factor 4H (eIF4H) gene [10]. Several studies have examined the role of miR-590-3p in cancer development, and both tumor-promoting and tumor-suppressing effects have been reported. While miR-590-3p was found to increase proliferation, migration, and/or invasion in T-cell acute lymphoblastic leukemia [11], colorectal cancer [12], and gastric cancer [13], it has also been reported to exert tumor-suppressive effects in breast cancer [14], liver cancer [15], glioblastoma [16], and bone cancer [17]. Recently, we identified tumor-promoting effects of miR590-3p on EOC cells. We found that miR-590-3p levels were upregulated in high-grade carcinoma and in the plasma of EOC patients, and that miR-590-3p enhanced cell proliferation, migration, invasion, and colony formation *in vitro*, and promoted tumor growth and metastasis *in vivo*. We further identified Forkhead box A2 (FOXA2) as a direct target for miR-590-3p [18]. However, additional target genes that may be involved in ovarian cancer development remain to be identified.

Cyclin G2 (CCNG2), a protein encoded by the *CCNG2* gene, belongs to a group of unconventional cyclins that play a role in maintaining cellular quiescence and cell cycle arrest [19,20], and has tumor suppressive effects in EOC [21]. It has been observed that CCNG2 levels are inversely correlated with cancer progression in breast [22], thyroid [23], oral [24], and gastric cancers [25], and in acute leukemia [26]. Previous studies from our lab have shown that CCNG2 reduced cell proliferation and was highly unstable in EOC cells [27,28,29]. More recently, we found that CCNG2 inhibited epithelial-to-mesenchymal transition, cell migration, invasion, and spheroid formation *in vitro*, as well as tumor formation and metastasis in vivo. We also identified that a major mechanism by which CCNG2 exerted these anti-tumor effects was via disrupting Wnt/β-catenin signaling [21].

FOXO3 (also known as FOXO3A) is a member of the Forkhead box class O transcription factor family [30,31]. It is an important regulator of several key aspects of cancer development [32]. Loss of FOXO3 has been reported to be an early event in the development of high-grade serous ovarian cancer [33], and low levels of FOXO3 are associated with poor patient survival [34,35]. Inhibition of FOXO3 expression promotes EOC cell proliferation and tumor progression, suggesting that FOXO3 exerts anti-tumor effects [36,37]. We have previously demonstrated that FOXO3 increased *CCNG2* transcription in EOC cells [27].

The Wnt/β-catenin pathway is tightly regulated, and aberration of this control often plays an important role in the development of many types of cancer, including EOC [38]. In the absence of Wnt ligands, β-catenin is phosphorylated in the cytoplasm and degraded by the destruction complex [38]. When activated by Wnt ligands, β-catenin is stabilized and translocated into the nucleus where it interacts with TCF/Lef transcription factors to regulate target gene expression [38]. The role of this pathway in promoting ovarian cancer stem cell self-renewal, chemoresistance, and metastasis has been reported [39,40].

To further investigate the mechanisms by which miR-590-3p promotes EOC development, we used bioinformatics tools to search for additional target genes and identified potential miR-590-3p binding sites on CCNG2 and FOXO3 3′ UTRs. Through reporter and functional assays, we demonstrated that miR-590-3p targeted CCNG2 and FOXO3 to promote proliferation, invasion, and formation of compact spheroids. We also found that miR-590-3p enhanced β-catenin activity.

## 2. Results

### 2.1. miR-590-3p Targets CCNG2

The 3′ UTR of CCNG2 harbours three predicted targeting sites for miR-590-3p (Figure 1A left). To determine if *CCNG2* is a target gene of miR-590-3p, a CCNG2 3′ UTR fragment containing the predicted miR-590-3p target sites was cloned into the pMIR-REPORT expression vector, downstream of the luciferase coding sequence. ES-2 cells were transfected with the reporter construct, together with miR-590-3p or a non-targeting control oligo (NC). Luciferase assays revealed that miR-590-3p decreased the luciferase activity when compared to NC (Figure 1A right).

To determine if miR-590-3p regulated CCNG2 expression, SKOV3.ip1 and ES-2 cells were transiently transfected with miR-590-3p or NC. Total RNA was extracted and reverse-transcribed. As revealed by qRT-PCR, CCNG2 mRNA levels were significantly lower in cells transfected with miR-590-3p than in the NC group (Figure 1B). In addition, we also used an ES-2 cell line that has been stably transfected with mir-590, which is the precursor of miR-590-3p and miR-590-5p, for functional studies. The cell line has been characterized previously and miR-590-3p was found to have tumor-promoting effects [18]. In the ES-2 cells stably transfected with mir-590, CCNG2 mRNA levels were significantly reduced when compared to cells expressing the control empty vector (EV) (Figure 1C, left). In addition, Western blot analyses revealed that mir-590 downregulated CCNG2 protein levels (Figure 1C, right).

### 2.2. miR-590-3p Targets FOXO3

A potential miR-590-3p binding site was identified in the 3′ UTR of FOXO3. Using a luciferase reporter assay, we found that miR-590-3p inhibited the luciferase activity of a construct containing FOXO3 3′ UTR (Figure 2A). The transient transfection of miR-590-3p also reduced the protein level of FOXO3 (Figure 2B). In the SKOV3.ip1 and ES-2 cells that stably overexpress mir-590, both the mRNA and protein levels of FOXO3 were lower than in the control cells (Figure 2C).

Since we have previously reported that FOXO3 induced CCNG2 transcription [27], we used a siRNA to silence FOXO3 expression and measured CCNG2 mRNA levels in siFOXO3-transfected EV and mir-590 stable cells. Knockdown of FOXO3 strongly decreased FOXO3 (Figure 2D) and CCNG2 (Figure 2E) mRNA levels.

### 2.3. Silencing of CCNG2 or FOXO3 Mimics the Effect of miR-590-3p

To determine if the inhibition of CCNG2 or FOXO3 expression had similar tumor-promoting effects as miR-590-3p overexpression, cells were transiently transfected with NC, siCCNG2, siFOXO3, or miR-590-3p, and various functional assays were performed. qRT-PCR and Western blotting confirmed that siCCNG2 and siFOXO3 decreased CCNG2 and FOXO3 expression, respectively, at both the mRNA (Figure 2D and Figure 3A) and protein (Figure 3B) levels. In cells transfected with siCCNG2, siFOXO3, or miR-590-3p, cell proliferation (Figure 3C), migration (Figure 3D), and invasion (Figure 3E) were all significantly increased when compared to NC.

### 2.4. Overexpression of CCNG2 or FOXO3 Reversed the Effect of miR-590-3p

To further confirm that CCNG2 and FOXO3 mediate the effect of miR-590-3p, rescue experiments were performed. SKOV3.ip1 cells were transfected with plasmids expressing FOXO3, Flag-tagged CCNG2, or their corresponding empty vectors, and overexpression of FOXO3 and CCNG2 was confirmed by Western blotting (Figure 4A). To determine if overexpression of CCNG2 would attenuate the effect of miR-590-3p, we transfected a miR-590-3p mimic or its control into both a stable cell line overexpressing CCNG2 and a control cell line expressing the empty vector [21]. miR-590-3p significantly increased the cell numbers in the control cells. However, in cells overexpressing CCNG2 the effect of miR-590-3p was strongly reduced (Figure 4B). In addition, we transiently transfected a CCNG2-expressing plasmid into mir-590-stable cells and measured cell invasion. As shown in Figure 4C, overexpression of CCNG2 attenuated the effect of mir-590 on cell invasion. Similarly, overexpression of FOXO3 significantly reduced the stimulatory effects of mir-590 on cell proliferation (Figure 4D) and invasion (Figure 4E).

### 2.5. miR-590-3p Induces the Formation of Compact Spheroids

Cancer cells possess varying capacities for spheroid formation [7]. It has been demonstrated that the more aggressive EOC cells form tighter and more compact spheroids [7]. We have previously reported that CCNG2 disrupts the formation of compact spheroids in 3D hanging drop culture [21]. To determine if miR-590-3p enhanced the spheroid formation, miR-590-3p or anti-miR-590-3p were transfected into SKOV3.ip1 or ES-2 cells and hanging drop spheroid formation assays were performed. Cells transfected with miR-590-3p formed smaller and more compact spheroids compared to the ones transfected with NC (Figure 5A). However, transfection of anti-miR-590-3p led to the formation of looser and larger spheroids (Figure 5B). To investigate if down-regulation of the miR-590-3p target genes mediated the effect of miR-590-3p on spheroid formation, stable cell lines overexpressing CCNG2 and its control were transfected with miR-590-3p or NC. The miR-590-3p transfected cells formed compact spheroids, however this effect was reduced in cells overexpressing CCNG2 (Figure 5C). The mir-590 stable cells were also transfected with either FOXO3, FOXA2, or their corresponding control plasmid. The mir-590 cells formed compact spheroids. Although the transfection of FOXO3 did not have a significant effect on the size of the spheroids, the spheroids formed by FOXO3-overexpressing cells appeared less dense than the control ones (Figure 5D). The overexpression of FOXA2 reduced the ability of the control and mir-590 cells to form spheroids, as indicated by the less dense and larger spheroids formed by FOXA2 transfected cells (Figure 5E).

### 2.6. mir-590 Enhances β-Catenin Signaling

Since CCNG2 has been found to inhibit β-catenin activity [21], we investigated whether miR-590-3p promoted β-catenin signaling. TOPFlash reporter assays were first performed to determine the transcriptional activity of β-catenin/TCF. In SKOV3.ip1 cells, the stable overexpression of mir-590 resulted in a significant increase in TOPFlash activity (Figure 6A). In addition, cell fractionation (Figure 6B) and confocal microscopy (Figure 6C) showed that more β-catenin accumulated in the nucleus of the mir-590 overexpressing cells when compared to the control cells. Finally, to determine if activation of the β-catenin pathway contributed to the tumor-promoting effects of miR-590-3p, SKOV3.ip1 cells were transiently transfected with a siRNA-targeting CTNNB1, the gene encoding β-catenin. Western blot and qRT-PCR confirmed that siCTNNB1 decreased both the protein and mRNA levels of β-catenin (Figure 6D). Silencing of *CTNNB1* expression in the mir-590 stable cells attenuated the ability of mir-590 cells to form tight spheroids (Figure 6E).

## 3. Discussion

We have recently found that miR-590-3p increased cell proliferation, migration, and invasion in vitro, and promoted tumor formation and metastasis in mouse xenografts models [18]. These findings, together with the higher expression levels of miR-590-3p in higher grade EOC tumors, suggest that miR-590-3p increases the aggressiveness of the EOC tumors. Consistent with this notion, in the present study we showed that miR-590-3p enhanced the formation of compact spheroids.

Recently, we reported that miR-590-3p promoted EOC development via a novel FOXA2-versican pathway. Specifically, we found that miR-590-3p directly targeted FOXA2 to inhibit its expression, resulting in the upregulation of versican levels [18]. In the present study, we identified CCNG2 and FOXO3 as additional target genes of miR-590-3p. First, luciferase reporter assays revealed that miR-590-3p targeted CCNG2 and FOXO3 3′ UTRs. Second, transient and/or stable overexpression of miR-590-3p decreased the mRNA and protein levels of CCNG2 and FOXO3. Finally, silencing of CCNG2 and FOXO3 mimicked, while their overexpression reduced, the effects of miR-590-3p on cell proliferation, migration, and/or invasion. The overexpression of CCNG2 and, to a lesser degree, FOXO3 also reversed the effect of miR-590-3p or mir-590 on compact spheroid formation. Similarly, the overexpression of FOXA2 reversed the effects of miR-590-3p on cell proliferation, migration, invasion [18], and spheroid formation (this study). These findings demonstrate that down-regulation of CCNG2, FOXO3, and FOXA2 all contribute to the tumor-promoting effects of miR-590-3p.

While this study is the first on EOC cells to show that CCNG2 is targeted by a miRNA, several studies on other types of cancer have reported that miRNAs can target CCNG2 to promote cancer development. For example, miR-1246 induced cell proliferation and invasion, as well as drug resistance, by targeting CCNG2 in breast cancer [41] and pancreatic cancer [42]. miR-135b has also been reported to target CCNG2 and exert tumor-promoting effects in lung cancer [43]. In addition, miR-340 promoted gastric cancer development by inhibiting CCNG2 [44]. Furthermore, miR-93 targeted CCNG2 to exert tumor-promoting effects in laryngeal squamous cell carcinoma [45]. Finally, a recent study revealed that miR-135A1 inhibited CCNG2 to promote colorectal cancer cell proliferation [46]. These studies reveal that CCNG2 is tightly regulated by miRNAs and that the down-regulation of CCNG2 by miRNAs is common to the development of many types of cancer.

FOXO3 is widely reported to have tumor-suppressive effects [36,37]. In this study, we found that silencing of FOXO3 stimulated cell proliferation, migration, and invasion, while the overexpression of FOXO3 inhibited these processes. These findings are consistent with previous reports on EOC cells. For example, FOXO3 has been shown to enhance drug sensitivity [34,36], and to suppress the self-renewal [47] and proliferation [37] of cancer stem-like cells. Recently, miR-551b was reported to promote ovarian cancer stem cell proliferation, invasion, and drug resistance, in part by suppressing FOXO3 expression [36]. Consistent with our previous findings that FOXO3 induced CCNG2 transcription [27], we found that silencing of FOXO3 strongly reduced CCNG2 mRNA levels. Therefore, it is likely that FOXO3 exerts anti-tumor effects on EOC cells in part by inducing CCNG2 expression. However, FOXO3 has been reported to regulate many genes involved in cancer development, such as p21 [48], ATR [49], ΔNp63 [50], and VEGF [51]. It is possible that other target genes of FOXO3 also contribute to the anti-tumor effects of FOXO3 in EOC.

In this study, we showed that cells overexpressing miR-590-3p formed more compact and much smaller spheroids than the control cells, while anti-miR-590-3p disrupted cell aggregation and compaction. Previous studies have revealed that more aggressive EOC cells form more compact spheroids [7]. These findings, together with our previous studies, which showed that miR-590-3p levels increased in high grade cancers and that miR-590-3p promoted EOC cell proliferation, migration, and invasion in vitro and tumor growth and metastasis in vivo, strongly supported the notion that miR-590-3p enhances the aggressiveness of EOC.

The Wnt/β-catenin signaling plays critical roles in the development of various types of cancers [38], including EOC [40]. We previously showed that CCNG2 overexpression decreased the levels of both DVL2 and LRP6 in EOC cells, and, hence, indirectly promoted β-catenin phosphorylation and degradation [21]. In this study, we found that miR-590-3p increased β-catenin accumulation in the nucleus and enhanced the transcriptional activity of β-catenin/TCF. Furthermore, we found that in mir-590 stable cells, knockdown of β-catenin by siRNA blocked the ability of miR-590-3p to promote the formation of tight spheroids. These results suggest that activation of β-catenin contributes to the tumor-promoting effects of miR-590-3p. The ability of miR-590-3p to promote β-catenin signaling is likely due to its down-regulation of CCNG2 and FOXO3, which is a transcriptional activator of CCNG2. In addition, FOXA2 has also been reported to inhibit β-catenin [52]. Therefore, down-regulation of FOXA2 by miR-590-3p may also lead to higher β-catenin activity. Recently, DKK1, an antagonist of the Wnt/β-catenin pathway, has been reported to be a miR-590-3p target gene [53]. In ES-2 cells stably transfected with mir-590, we found that DKK1 levels were strongly down-regulated [18]. Therefore, it is possible that miR-590-3p also targets DKK1 to enhance β-catenin signaling and this will be investigated in the future.

In summary, we provided further evidence to support a tumor-promoting role of miR-590-3p in EOC. We also revealed additional targets of miR-590-3p and identified β-catenin as a key pathway mediating the effects of miR-590-3p on EOC cells. Based on our findings, we propose that miR-590-3p targets FOXA2, CCNG2, FOXO3, and, probably, DKK1 in EOC cells. Through down-regulation of these genes, miR-590-3p increased β-catenin activity to promote EOC development (Figure 7). These findings provide further insights into the molecular mechanisms underlying the carcinogenesis of ovarian cancer.

## 4. Materials and Methods

### 4.1. Cell Culture

ES-2 cells were purchased from American Type Culture Collection (ATCC, Manassas, VA, USA), while SKOV3.ip1 cells were generously donated by Dr. Mien-Chie Hung (University of Texas, M.D. Anderson Cancer Center, Huston, TX, USA), as previously reported [21]. These cells were maintained in McCoy 5A media (Sigma-Aldrich, Oakville, ON, Canada) and supplemented with 10% Fetal Bovine Serum (FBS), 100 IU/mL penicillin, and 100μg/mL streptomycin (all purchased from Life Technologies, Thermo Fisher Scientific, Mississauga, ON, Canada). Cells were cultured in a humidified atmosphere of 5% CO_2_ at 37 °C.

### 4.2. Transient Transfection

Transient transfection of plasmids (0.25 µg), miRNA mimics, inhibitors, or siRNA (150–200 nM) were carried out using Lipofectamine 2000 or Lipofectamine RNAiMAX (Life Technologies), following the manufacturer’s suggested procedures. Non-targeting negative control (NC) (5′-UUCUCCGAACGUGUCACGU-3′), hsa-miR-590-3p mimic (5′-UAAUUUUAUGUAUAAGCUAGU -3′) [18], siCCNG2 (5′-AUGCCUAGCCGAGUAUUCU-3′) [21], siCTNNB1 (5′-AAUGCUUGGUUCACCAGUGGA-3′), and siFOXO3 (5′-CAACCUGUCACUGCAUAGU-3′) were purchased from GenePharma Co. (Shanghai, China). Anti-miR-590-3p and its corresponding NC were purchased from RiboBio (Guangzhou, China). Transient transfection was carried out for 6 h in Opti-MEM Reduced Serum Medium (Thermo Fisher Scientific), followed by 18 h of recovery in media containing 10% FBS.

### 4.3. Constructs and Stable Cell Lines

CCNG2 was cloned into either a retroviral vector, pBabe-puro [21], or a plasmid vector, p3XFLAG-CMV-7.1 [28], for stable and transient transfection, as previously reported. Human FOXO3 plasmid [54] was obtained from Addgene (Cambridge, MA, USA). ES-2 and SKOV3.ip1 cells stably overexpressing the mir-590 precursor stem-loop sequence were cloned into the pRNAT-CMV3.2/Hygro expression vector (GenScript). Control cells (Empty Vector, EV), were transfected with pRNAT-CMV3.2/Hygro, without the mir-590 insert, as previously described [18]. Control and CCNG2 stable cells were generated and validated, as described previously [21].

### 4.4. RNA Extraction and Real-Time PCR

Total RNA was isolated using TRIzol reagent (Invitrogen, Thermo Fisher Scientific), as previously described [55]. To determine mRNA levels, 1.5 μg of total RNA was used to synthesize first strand cDNA by M-MuLV reverse transcriptase (New England BioLabs Ltd., Whitby, ON, Canada). Real-time PCR was carried out using gene-specific primers and EvaGreen qPCR Master Mix (Applied Biological Materials, Richmond, BC, Canada). The expression levels of mRNA were normalized to GAPDH. The relative expression levels of mRNAs were determined using the standard 2^−ΔΔ*C*t^ method.

### 4.5. Protein Extraction and Immunoblotting

Cells were lysed in a RIPA buffer (20 mM Tris, pH 8.0, 150 mM NaCl, 10 mm NaF, 0.1% SDS, 1% Nonidet P-40, and 1× protease inhibitor cocktail) (Pierce, IL, USA). Cell lysates were then collected by centrifugation at 12,000× *g* for 20 min at 4 °C. A bicinchoninic acid (BCA) assay was conducted to measure protein concentration. Equal amounts of protein were subjected to 10% SDS-polyacrylamide gel electrophoresis and transferred onto polyvinylidene difluoride (PVDF, Bio-Rad) membranes. The membranes were then blocked with 5% milk for 1 h at room temperature followed by an incubation with primary antibodies in milk overnight at 4 °C. The membranes were washed with TBS-T and subsequently probed with an HRP-conjugated secondary antibody (1:5000) at room temperature for 2 h. Signals were visualized using ECL (Millipore or Bio-Rad, Canada) according to the manufacturers’ protocols. CCNG2 antibody was obtained from Abcam (1:500). β-catenin and FOXA2 antibodies were obtained from Cell Signaling and diluted to 1:1000 and 1:2000, respectively. FOXO3 (1:1000), Lamin B (1:2000), and GAPDH (1:10,000) antibodies were purchased from Santa Cruz.

### 4.6. Luciferase Assays

To confirm that miR-590-3p targeted CCNG2 and FOXO3, luciferase reporter plasmids in which a fragment of the CCNG2 or FOXO3 3′ UTR was cloned into the pMIR-REPORT (Ambion, Austin, TX, USA) downstream of the luciferase coding sequence were constructed. The 3′ UTR fragments were generated by PCR using forward primers, 5′-GATAGTCTAGTCATTGCATG-3′, and reverse primers, 5′-GCGTGCACCACATCCTAGAA-3′, for CCNG2, and forward primers, 5′-TGAATGATTGGTCATGAGGC-3′, and reverse primers, 5′-AACATGTGAAGCCAAAATGC-3′, for FOXO3. To assess β-catenin-TCF/LEF transcriptional activity, the TOPFlash construct [56], obtained from Addgene (Cambridge, MA, USA), was used.

Cells were seeded in 12-well plates at a density of 75 × 10^4^ cells/well, and transfected with a pMIR-Report or TOPFlash plasmid and the pRL-TK internal control (encoding Renilla luciferase) plasmid. Five hours after transfection, cells were recovered in media supplemented with 10% FBS. Twenty-four hours after transfection, the cells were lysed and luciferase activities were measured using the Dual-Luciferase Reporter Assay System (Promega, Madison, WI, USA) according to the manufacturer’s instructions, as previously reported [57].

### 4.7. Spheroid Formation Assay (3D Hanging Drop)

Spheroid formation assays were performed as described previously [7,21]. Briefly, at 24 h after transfection, cells were collected and 2 × 10^4^ cells were re-suspended in 20 μL of culture media. Between 10 and 21 drops of cells were placed on the inside of a 100 mm culture dish cover for 3–4 days and spheroids were photographed. The circumferences of spheroids were measured using the ImageJ Software.

### 4.8. Migration and Invasion Assays

Migration and invasion assays were performed, as previously described [18]. Briefly, for migration experiments, cells were seeded onto Transwell membranes (Corning, Massachusetts, USA) at a density of 12 × 10^3^ cells per well. To assess invasion, cells were seeded onto Transwell membranes pre-coated with a thin layer of Matrigel (R&D Systems, MN, USA), at a density of 15 × 10^3^. At 24 h after seeding, cells were fixed and stained. Cells on the top of the Transwell were removed by swabbing and filters were cut and mounted on a slide. Pictures were taken for the whole fields and cells were counted using an automated quantification plugin for ImageJ [58].

### 4.9. Immunofluorescence

Cells were seeded on the top of coverslips and cultured in serum-free conditions for 18 h, followed by incubation with media containing 10% FBS for 6 h. They were fixed with Acetone/Methanol (1:1) for 15 min at −20 °C and blocked with 1% BSA for 1 h at room temperature. Cells were incubated with an anti-β-catenin antibody (Cell Signaling) and diluted (1:100) in 1% BSA overnight, followed by incubation with Alexa Fluor^®^ 488 secondary antibody (Invitrogen) for 2 h. Coverslips were then incubated with 4,6-diamidino-2-phenylindole (DAPI, 1 μg/mL) for 10 min. Fluorescent signals were observed using the Zeiss LSM 700 confocal microscope (Toronto, ON, Canada).

### 4.10. Statistical Analysis

All experiments were done at least three times with at least triplicates in each group. The results are expressed as the mean ± SEM in bar graphs. GraphPad Prism 6 was used to perform all statistical analyses. Comparison between two groups was performed using Student’s *t*-test, while multiple groups were analyzed by one-way or two-way ANOVA followed by Student-Newman-Keuls post hoc test. Significance was defined as *p* < 0.05.

## Figures and Tables

**Figure 1 ijms-20-01810-f001:**
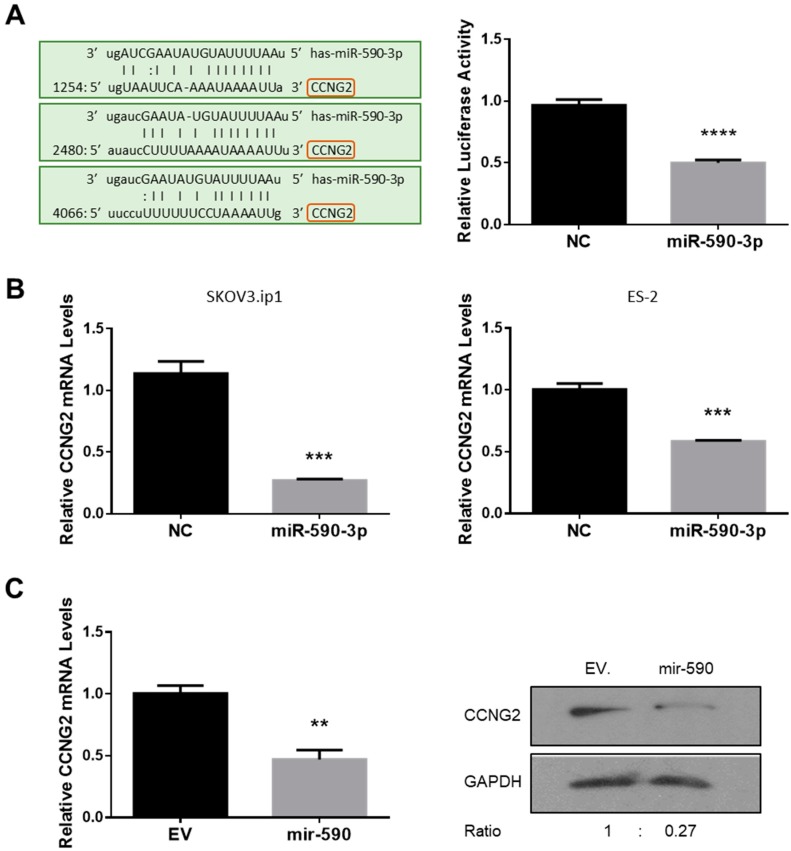
*CCNG2* is a target gene of miR-590-3p. (**A**) miR-590-3p targets CCNG2 3ʹ UTR. Three predicted miR-590-3p binding sites were found in CCNG2 3ʹ UTR. A luciferase reporter construct containing CCNG2 3ʹ UTR was generated using the pMIR-REPORT vector. Transient transfection of miR-590-3p significantly decreased the luciferase activity in ES-2 cells when compared to cells transfected with a non-targeting oligonucleotide (NC). (**B**) miR-590-3p downregulated CCNG2 mRNA levels. In ES-2 and SKOV3.ip1 cells transiently transfected with miR-590-3p, the CCNG2 mRNA levels were significantly lower than in the control cells transfected with NC. (**C**) mir-590 inhibited CCNG2 expression. CCNG2 mRNA (left) and protein (right) levels were lower in the ES-2 cells stably overexpressing miR-590-3p precursor, mir-590, than in the control cells expressing the empty vector (EV). Data represent the mean ± SEM (*n* = 3). Statistical analyses were performed using Student’s *t*-test. ** *p* < 0.01, *** *p* < 0.001, **** *p* < 0.0001 vs. controls.

**Figure 2 ijms-20-01810-f002:**
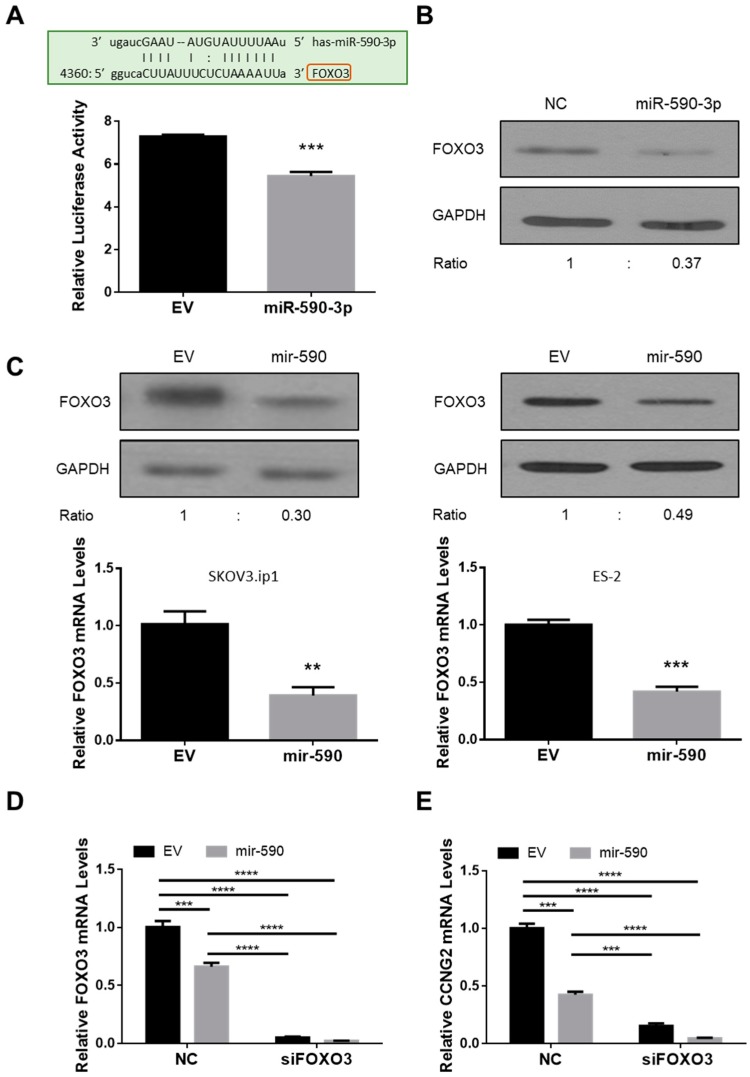
miR-590-3p targeted *FOXO3*. (**A**) A predicted miR-590-3p binding site was found in FOXO3 3ʹ UTR (top). A luciferase reporter assay showed that mir-590-3p significantly decreased the luciferase activity in ES-2 cells (bottom). (**B**) miR-590-3p decreased the protein level of FOXO3 in SKOV3.ip1 cells. (**C**) In SKOV3.ip1 and ES-2 cells stably transfected with mir-590, both the mRNA and protein levels of FOXO3 were lower than control cells transfected with an empty vector (EV). (**D**) ES-2 cells were transfected with siRNA targeting FOXO3. In siFOXO3 transfected cells, FOXO3 mRNA levels were significantly down-regulated. **(E)** Silencing of FOXO3 reduced CCNG2 mRNA level in ES-2. Data represent the mean ± SEM (*n* = 3). Statistical analyses were performed using Student’s t-test (A and C), or two-way ANOVA/Student–Newman–Keuls test (D and E). ** *p* < 0.01, *** *p* < 0.001, **** *p* < 0.0001.

**Figure 3 ijms-20-01810-f003:**
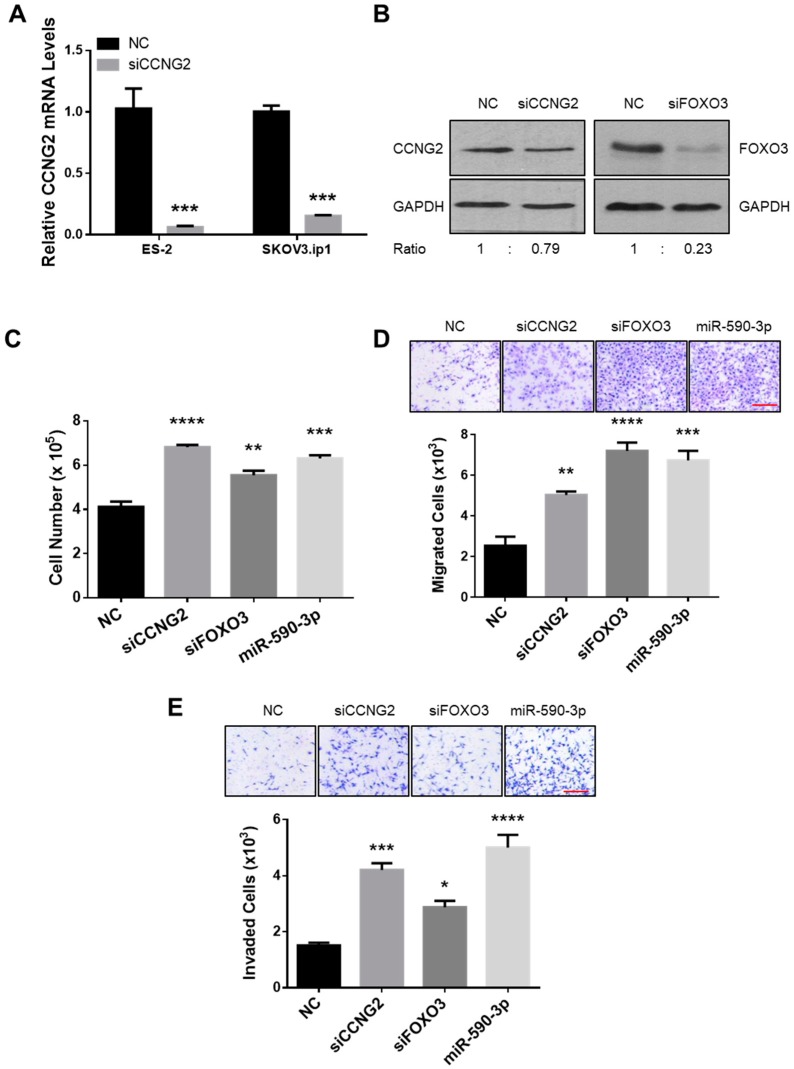
Silencing of CCNG2 or FOXO3 mimicked the effects of miR-590-3p. (**A**) ES-2 and SKOV3.ip1 cells were transiently transfected with a siRNA-targeting CCNG2 and a non-targeting oligonucleotide (NC). Real-time PCR confirmed the down-regulation of CCNG2 mRNA after siCCNG2 transfection. (**B**) Transient transfection of siCCNG2 and siFOXO3 into ES-2 cells reduced the protein levels of CCNG2 and FOXO3, respectively. (**C**) ES-2 cells were transiently transfected with siCCNG2, siFOXO3, miR-590-3p, or NC, and cell numbers were determined at 48 h after transfection. (**D**) SKOV3.ip1 cells were transiently transfected with siCCNG2, siFOXO3, miR-590-3p, or NC, and cell migration was determined. siCCNG2, siFOXO3, and miR-590-3p all significantly increased the number of migrated cells when compared to NC. (**E**) ES-2 cells were transiently transfected with siCCNG2, siFOXO3, miR-590-3p, or NC. Silencing of CCNG2 or FOXO3, or the overexpression of miR-590-3p induced cell invasion. Red scale bar (D and E) = 25µm. Data represent the mean ± SEM (*n* = 3). Statistical analyses were performed using Student’s *t*-test (A), or one-way ANOVA/Student–Newman–Keuls test (C, D, and E). * *p* < 0.05, ** *p* < 0.01, *** *p* < 0.001, **** *p* < 0.0001.

**Figure 4 ijms-20-01810-f004:**
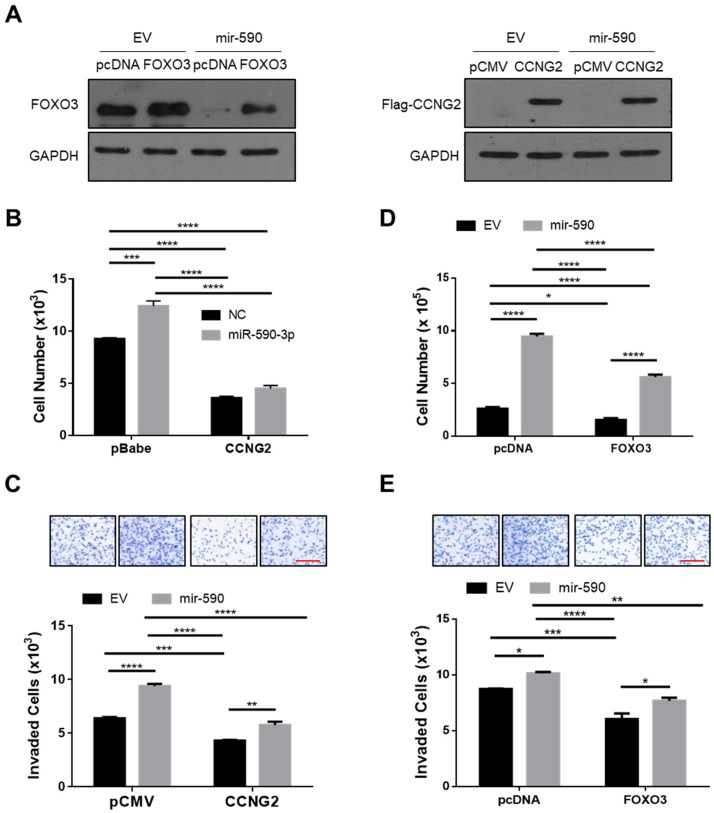
Overexpression of CCNG2 and FOXO3 reversed the effects of miR-590-3p on cell proliferation and invasion. (**A**) Confirmation of FOXO3 and CCNG2 overexpression. SKOV3.ip1 cells stably overexpressing mir-590 or its empty vector (EV) were transiently transfected with FOXO3 (left), CCNG2 (right), or their corresponding control vectors. Western blot analyses were performed using FOXO3 or Flag (to detect CCNG2) antibody. (**B**) Overexpression of CCNG2 attenuated the effect of miR-590-3p on cell proliferation. SKOV3.ip1 cells stably transfected with CCNG2 or its control vector pBabe-puro were transiently transfected with either miR-590-3p or a non-targeting control oligo (NC). The effect of miR-590-3p was blocked by CCNG2 overexpression. (**C**) The overexpression of CCNG2 reduced the effect of mir-590 on cell invasion. ES-2 cells stably expressing mir-590 or its empty vector (EV) were transiently transfected with either CCNG2 or its empty vector pCMV, and cell invasion was determined. (**D**) Transient transfection of FOXO3 into SKOV3.ip1 stably expressing mir-590 or EV decreased cell proliferation and (**E**) cell invasion. Red scale bar (C and E) = 25µm. Data represent the mean ± SEM (*n* = 3). Statistical analyses were performed using two-way ANOVA/Student–Newman–Keuls test. * *p* < 0.05, ** *p* < 0.01, *** *p* < 0.001, **** *p* < 0.0001.

**Figure 5 ijms-20-01810-f005:**
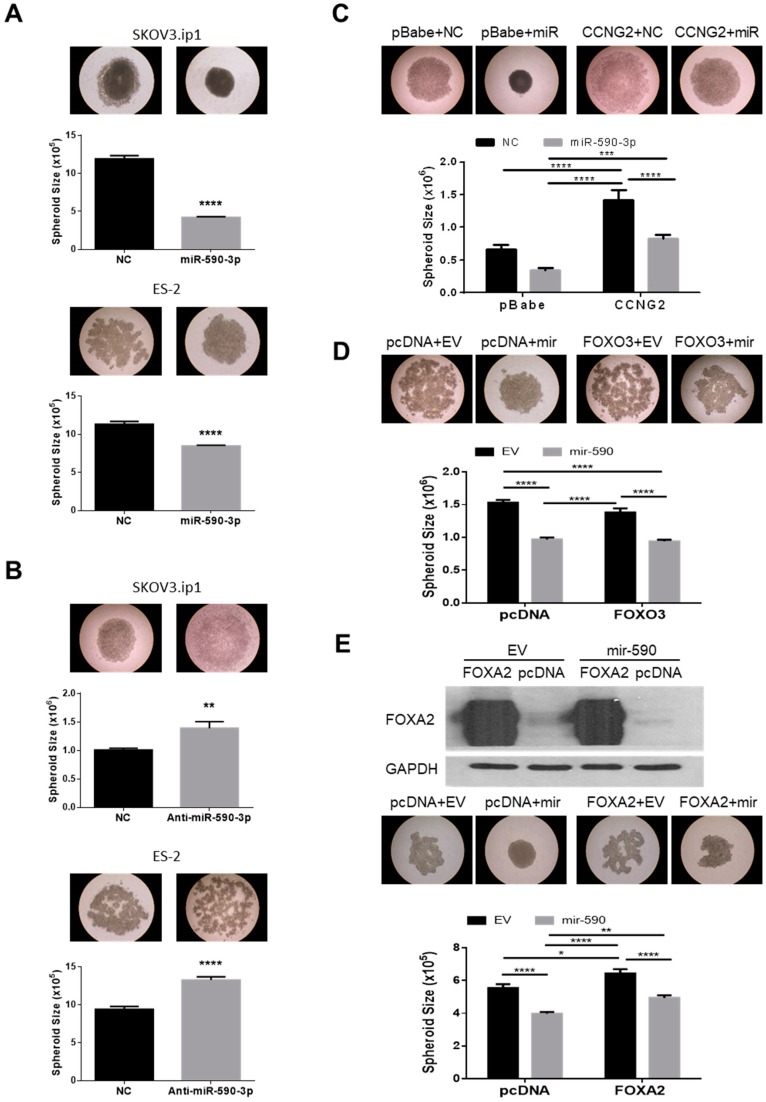
miR-590-3p promoted the formation of compact spheroids. (**A**) The overexpression of miR-590-3p induced compact spheroid formation in hanging drop culture. Transient transfection of miR-590-3p into both SKOV3.ip1 (top) and ES-2 (bottom) cells resulted in the formation of tight, compact, and smaller spheroids when compared to the non-targeting control (NC) (*n* = 12–21). (**B**) Inhibition of miR-590-3p blocked compact spheroid formation. Transient transfection of anti-miR-590-3p disrupted the formation of compact spheroids (*n* = 18–21). (**C**) Overexpression of CCNG2 reversed the effect of miR-590-3p in spheroid formation. Control or CCNG2 stable SKOV3.ip1 cells were transiently transfected with either miR-590-3p or NC, and hanging drop cultures were performed. In control cells, miR-590-3p overexpression resulted in the formation of compact spheroids. CCNG2 stable cells formed loose spheroids and reduced the effect of miR-590-3p on spheroid formation (*n* = 12–15). (**D**) Control and mir-590 stable ES-2 cells were transiently transfected with either FOXO3 or its empty vector, pcDNA3. FOXO3 overexpression partially inhibited the ability of mir-590 to form the compact spheroids, but had no effect on the size of the spheroids (*n* = 16–20). (**E**) FOXA2 reduced the effect of mir-590 on spheroid formation. Transfection of FOXA2 into control and mir-590-overexpressing SKOV3.ip1 resulted in the formation of larger and less compact spheroids (bottom) (*n* = 16–20). Overexpression of FOXA2 after transfection was confirmed by Western blotting (top). Data represent the mean ± SEM. Statistical analyses were performed using *t*-test (A and B), or two-way ANOVA/Student–Newman–Keuls test. * *p* < 0.05, ** *p* < 0.01, *** *p* < 0.001, **** *p* < 0.0001.

**Figure 6 ijms-20-01810-f006:**
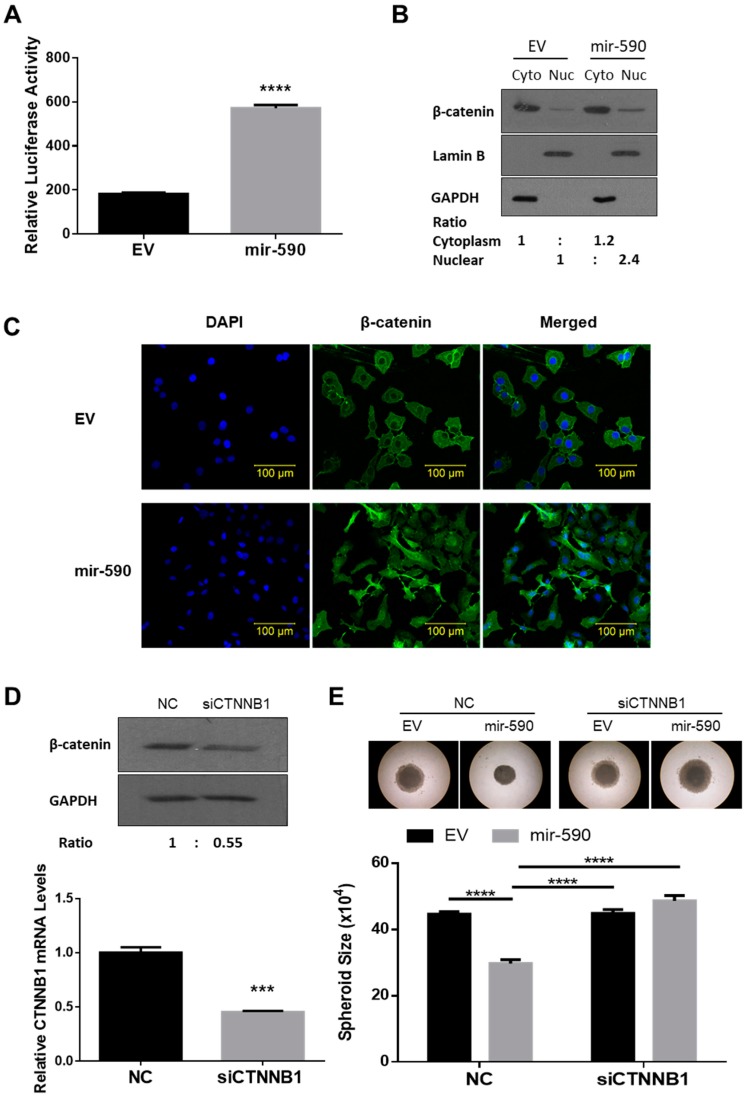
mir-590 enhanced β-catenin signaling. (**A**) A TOPFlash reporter assay was performed in SKOV3.ip1 cells stably transfected with mir-590 or its control empty vector (EV). mir-590 overexpression increased β-catenin transcriptional activity. Data represent the mean ± SEM (*n* = 3). (**B**) Cytosolic (Cyto) and nuclear (Nuc) fractions were prepared from EV and mir-590-overexpressing SKOV3.ip1 cells, and Western blotting was performed. mir-590 cells exhibited higher β-catenin in their nucleus. The densitometry readings of cytoplasmic and nuclear fractions were normalized to their corresponding GAPDH and Lamin B levels, respectively. (**C**) Immunofluorescence staining of β-catenin showed that SKOV3.ip1 cells stably overexpressing mir-590 had strong β-catenin signals in their nucleus. (**D**) Knockdown of CTNNB1 using siRNA reduced the mRNA and protein levels of β-catenin. (**E**) Silencing of CTNNB1 blocked the effect of mir-590 on spheroid formation in SKOV3.ip1 cells. Data represent the mean ± SEM (*n* = 10). Statistical analyses were performed using Student’s *t*-test (A and D), or two-way ANOVA/Student–Newman–Keuls test (E). *** *p* < 0.001, **** *p* < 0.0001.

**Figure 7 ijms-20-01810-f007:**
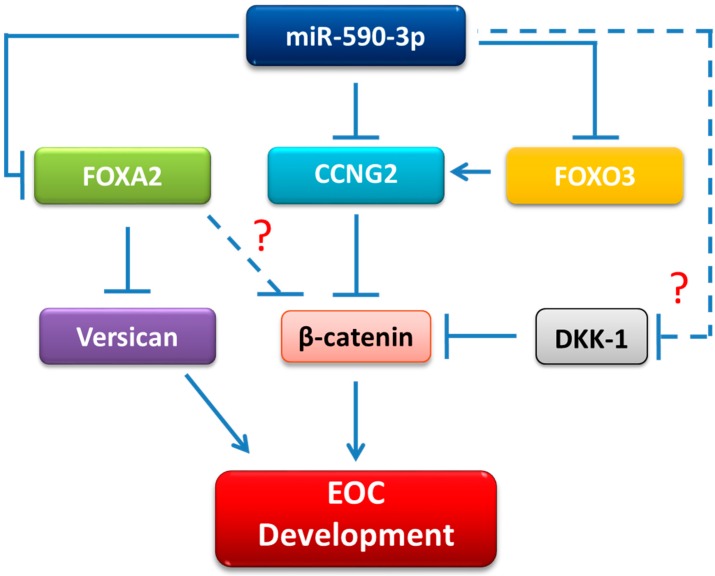
The proposed mechanism of mir-590 in promoting EOC development. miR-590-3p targeted the *CCNG2*, *FOXO3*, *FOXA2,* and, probably, *DKK1*, leading to an increase in the β-catenin activity. Down-regulation of FOXA2 also led to the upregulation of versican. The direct targeting of DKK1 by miR-590-3p in EOC cells remains to be validated. Whether or not FOXA2 inhibits β-catenin also requires further investigation. We propose that the up-regulation of β-catenin and versican is the key event underlying the tumor-promoting effects of miR-590-3p in EOC cells.

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
