# Peer review of "miR-590-3p Targets Cyclin G2 and FOXO3 to Promote Ovarian Cancer Cell Proliferation, Invasion, and Spheroid Formation"

_ijms, 2019, doi:10.3390/ijms20081810_

Round 1

Reviewer 1 Report

Overview

The Peng lab has reported on the function and signaling pathways affecting CCNG2 in ovarian cancer for many years. Their central thesis is that CCNG2 is a tumor suppressor and is downregulated in ovarian cancer. In recent work they identified the microRNA miR-590-3p as a tumor suppressor and suggested its suppressive effects may be mediated by FOXA2 and VCAN. In this manuscript, by Salem, et al., they propose another hypothesis that miR-590-3p is tumor suppressive due to its inhibitory effects on CCNG2 and FOXO3 mRNA. They provide evidence for this hypothesis by upregulating/downregulating miR-590-3p, CCNG2, FOXO3, and beta-catenin in ES-2 (a poorly differentiated clear cell ovarian cancer line) and SKOV3.ip1 (cystadenocarcinoma serous cell line derived from ascites from mouse injected with SKOV3 cells, with mutations in APC, FBXW7, PIK3CA and TP53).  They measure mRNA levels, protein levels, proliferation, invasion, migration, nuclear accumulation of B-cat, and hanging drop spheroid sizes in these cells. In general, the results of these experiments support their hypothesis.

Major Critiques

Lines 170-171: "The mir-590 cells formed compact spheroids and transfection with FOXO3 (Fig. 5D) or FOXA2 (Fig. 5E) reduced the effect of mir-590 on promoting compact spheroid formation." Figure 5D does not appear to support this claim.  In figure 5D it looks like there was no significant difference between spheroid size of either empty vector or mir-590 cells when transfected with FOXO3.  Either remove this claim or provide experimental data to support it.

The study relies on effects seen in ES-2 and SKOV3.ip1 cells. ES-2 cells were derived from clear cell ovarian cancer and SKOV3.ip1 cells were derived from ascites in immunocompromised mice injected with the parental SKOV3 line. SKOV3 cells are originally from ascites from a patient with serous cystadenocarcinoma, although reports from Domcke, et al., Nature Communications 2013, place doubt on their origin as high grade serous. This means the results may not be applicable to the most common form of ovarian cancer, high grade serous.  If the authors provided data from cell lines that are considered bona fide HGSOC lines, it would strengthen the manuscript. However, the findings reported are still valid and worthy of publication.

In previous papers, the Peng lab has provided data from mining human tumor databases like TCGA.  There is no equivalent in this paper, which reduces the significance, nor are there any in vivo models used.

Minor Critiques

Fig 3A Western blot shows very minor reduction in CCNG2 after treatment with siRNA. Please perform densitometry or repeat to define the level of reduction.

Fig 2D, 2E, 4A-D, 5C-E, 6E. Using letters to indicate significance is hard to understand.  Please change to the more standard method of drawing a line between the two bars being compared and use *, **, or *** to indicate significance between the two bars.

Fig 6B Western blot shows very minor increase in beta-catenin in nucleus, which is a central point of the manuscript. Please perform densitometry or repeat to define the level of reduction. (IF in 6C is more convincing).

Reference 17 and 26 are identical

Line 93: Last sentence is not continued on following page.

Author Response

Thank you for the comments and suggestions.  Below are our responses: 

Major Critiques

Lines 170-171: "The mir-590 cells formed compact spheroids and transfection with FOXO3 (Fig. 5D) or FOXA2 (Fig. 5E) reduced the effect of mir-590 on promoting compact spheroid formation." Figure 5D does not appear to support this claim.  In figure 5D it looks like there was no significant difference between spheroid size of either empty vector or mir-590 cells when transfected with FOXO3.  Either remove this claim or provide experimental data to support it.

Response: Although there are no significant changes in the size of spheroids after FOXO3 overexpression, the spheroids are less dense.  We have modified the statement in lines 247-251.

The study relies on effects seen in ES-2 and SKOV3.ip1 cells. ES-2 cells were derived from clear cell ovarian cancer and SKOV3.ip1 cells were derived from ascites in immunocompromised mice injected with the parental SKOV3 line. SKOV3 cells are originally from ascites from a patient with serous cystadenocarcinoma, although reports from Domcke, et al., Nature Communications 2013, place doubt on their origin as high grade serous. This means the results may not be applicable to the most common form of ovarian cancer, high grade serous.  If the authors provided data from cell lines that are considered bona fide HGSOC lines, it would strengthen the manuscript. However, the findings reported are still valid and worthy of publication.

Response: According to reference # 2, SKOV3 is now considered an endometrioid cell line. ES-2 has mixed features.  Our data show that miR-590-3p, FOXO3, and cyclin G2 are effective in these cells. However, they are insufficient for us to make a conclusion about whether their effects are specific to certain subtypes of EOC. In a previous study (reference 18), we suggest that the tumor-promoting effects of miR-590-3p are not limited to a specific subtype of EOC.  

In previous papers, the Peng lab has provided data from mining human tumor databases like TCGA.  There is no equivalent in this paper, which reduces the significance, nor are there any in vivo models used.

Response:   We have reported in vivo studies for both miR-590-3p and CCNG2.  Clinical relevance of miR-590-3p, CCNG2, and FOXA2 have also been reported in our previous publications (references 18 and 21).  Loss of FOXO3 expression in ovarian cancer has also been reported by other groups (references 31 and 32).  The major focus of the present study is to further uncover the molecular mechanisms by which miR-590-3p promotes ovarian cancer development.

Minor Critiques

Fig 3A Western blot shows very minor reduction in CCNG2 after treatment with siRNA. Please perform densitometry or repeat to define the level of reduction.

Response: We have performed densitometry for both CCNG2 siRNA and FOXO3 siRNA and the results are presented in Fig 3B. Since the CCNG2 antibody is no longer available, we repeated the experiment and measured CCNG2 mRNA level.  The data are now presented in Fig. 3A.  

Fig 2D, 2E, 4A-D, 5C-E, 6E. Using letters to indicate significance is hard to understand.  Please change to the more standard method of drawing a line between the two bars being compared and use *, **, or *** to indicate significance between the two bars.

Response:  In the previous version, we used letters to indicate statistical significance.  In this case, if two bars share the same letter, they are not significant from each other. This is a simple way to denote statistical significance.  We have made changes according to the suggestion.  However, the new figures look very busy with all the lines and symbols.

Fig 6B Western blot shows very minor increase in beta-catenin in nucleus, which is a central point of the manuscript. Please perform densitometry or repeat to define the level of reduction. (IF in 6C is more convincing).

Response: Densitometry reading is included in Fig. 6B. The nuclear beta-catenin level in mir-590 stable cells is more than 2 fold of that in the control cells.

Reference 17 and 26 are identical

Response: Corrected.

Line 93: Last sentence is not continued on following page.

Response: Corrected. 

Thank you for the comments and suggestions.  Below are our responses: 

Major Critiques

Lines 170-171: "The mir-590 cells formed compact spheroids and transfection with FOXO3 (Fig. 5D) or FOXA2 (Fig. 5E) reduced the effect of mir-590 on promoting compact spheroid formation." Figure 5D does not appear to support this claim.  In figure 5D it looks like there was no significant difference between spheroid size of either empty vector or mir-590 cells when transfected with FOXO3.  Either remove this claim or provide experimental data to support it.

Response: Although there are no significant changes in the size of spheroids after FOXO3 overexpression, the spheroids are less dense.  We have modified the statement in lines 247-251.

The study relies on effects seen in ES-2 and SKOV3.ip1 cells. ES-2 cells were derived from clear cell ovarian cancer and SKOV3.ip1 cells were derived from ascites in immunocompromised mice injected with the parental SKOV3 line. SKOV3 cells are originally from ascites from a patient with serous cystadenocarcinoma, although reports from Domcke, et al., Nature Communications 2013, place doubt on their origin as high grade serous. This means the results may not be applicable to the most common form of ovarian cancer, high grade serous.  If the authors provided data from cell lines that are considered bona fide HGSOC lines, it would strengthen the manuscript. However, the findings reported are still valid and worthy of publication.

Response: According to reference # 2, SKOV3 is now considered an endometrioid cell line. ES-2 has mixed features.  Our data show that miR-590-3p, FOXO3, and cyclin G2 are effective in these cells. However, they are insufficient for us to make a conclusion about whether their effects are specific to certain subtypes of EOC. In a previous study (reference 18), we suggest that the tumor-promoting effects of miR-590-3p are not limited to a specific subtype of EOC.  

In previous papers, the Peng lab has provided data from mining human tumor databases like TCGA.  There is no equivalent in this paper, which reduces the significance, nor are there any in vivo models used.

Response:   We have reported in vivo studies for both miR-590-3p and CCNG2.  Clinical relevance of miR-590-3p, CCNG2, and FOXA2 have also been reported in our previous publications (references 18 and 21).  Loss of FOXO3 expression in ovarian cancer has also been reported by other groups (references 31 and 32).  The major focus of the present study is to further uncover the molecular mechanisms by which miR-590-3p promotes ovarian cancer development.

Minor Critiques

Fig 3A Western blot shows very minor reduction in CCNG2 after treatment with siRNA. Please perform densitometry or repeat to define the level of reduction.

Response: We have performed densitometry for both CCNG2 siRNA and FOXO3 siRNA and the results are presented in Fig 3B. Since the CCNG2 antibody is no longer available, we repeated the experiment and measured CCNG2 mRNA level.  The data are now presented in Fig. 3A.  

Fig 2D, 2E, 4A-D, 5C-E, 6E. Using letters to indicate significance is hard to understand.  Please change to the more standard method of drawing a line between the two bars being compared and use *, **, or *** to indicate significance between the two bars.

Response:  In the previous version, we used letters to indicate statistical significance.  In this case, if two bars share the same letter, they are not significant from each other. This is a simple way to denote statistical significance.  We have made changes according to the suggestion.  However, the new figures look very busy with all the lines and symbols.

Fig 6B Western blot shows very minor increase in beta-catenin in nucleus, which is a central point of the manuscript. Please perform densitometry or repeat to define the level of reduction. (IF in 6C is more convincing).

Response: Densitometry reading is included in Fig. 6B. The nuclear beta-catenin level in mir-590 stable cells is more than 2 fold of that in the control cells.

Reference 17 and 26 are identical

Response: Corrected.

Line 93: Last sentence is not continued on following page.

Response: Corrected. 

Reviewer 2 Report

The manuscript title: miR-590-3ptargets cyclin G2 and FOXO3 to promote ovarian cancer cell proliferation, invasion and spheroid formation. is well written and with scientific interest in the ovarian cancer area. This manuscript present a good experimental design. In my opinion this manuscript would be published

Author Response

thank you.

Reviewer 3 Report

In this manuscript Salem et. al., present evidences to suggest that miRNA miR-590-3p targets Cyclin G2 and FOXO3 to promote Ovarian cancer. The authors first validate that, Cyclin G2 (CCNG2) and FOXO3 are real targets of miR-590. Further they provide data to show that knockdown of CCNG2 or FOXO3 have similar effects on cell proliferation, cell migration like miR-590 over-expression. Over-expression of CCNG2 could rescue the effect of miR-590-3p over-expression on cell proliferation and cell migration. Finally the authors show that, miR-590 brings out the effect on cancer cells mediated via β-catenin. This is an informative article, which provides new data on miR590 mediated cancer cell proliferation via CCNG2 and FOXO3. I have following concerns regarding this manuscript;

The authors have tried to join multiple dots in this article without giving enough evidences. For example, the authors show that miR-590 targets both CCNG2 and FOXO3. Further they refer that FOXO3 controls CCNG2 expression. Now its not clear if the effect of miR590 on cancer cells is only due to down regulation of CCNG2 or FOXO3 also plays some independent roles. The authors should try to rescue the FOXO3 knockdown effect by over-expressing CCNG2 to answer this conundrum. I have similar concerns for the β-catenin part too and I felt rescue experiments are necessary to join these dots. 

In fig. 3: Its difficult to understand why these is a difference in the assay readouts in these siRNA of miRNA transfection conditions. I think the trends should look similar in all the assays. 

Minor Comments;

Line 31: EOC full form missing.

Line 93: Sentence missing.

In all the figure legends, which statistical methods  have been used are missing. 

Line 112: Information regarding siRNA transfection conditions and timings missing in the methods section. 

Fig. 3A: The right top panel of the blot is of FOXO3?

Significance values are missing in many of the graphs. 

Author Response

Reviewer 3

In this manuscript Salem et. al., present evidences to suggest that miRNA miR-590-3p targets Cyclin G2 and FOXO3 to promote Ovarian cancer. The authors first validate that, Cyclin G2 (CCNG2) and FOXO3 are real targets of miR-590. Further they provide data to show that knockdown of CCNG2 or FOXO3 have similar effects on cell proliferation, cell migration like miR-590 over-expression. Over-expression of CCNG2 could rescue the effect of miR-590-3p over-expression on cell proliferation and cell migration. Finally the authors show that, miR-590 brings out the effect on cancer cells mediated via β-catenin. This is an informative article, which provides new data on miR590 mediated cancer cell proliferation via CCNG2 and FOXO3. I have following concerns regarding this manuscript; The authors have tried to join multiple dots in this article without giving enough evidences. For example, the authors show that miR-590 targets both CCNG2 and FOXO3. Further they refer that FOXO3 controls CCNG2 expression.

Now its not clear if the effect of miR590 on cancer cells is only due to down regulation of CCNG2 or FOXO3 also plays some independent roles.

The authors should try to rescue the FOXO3 knockdown effect by over-expressing CCNG2 to answer this conundrum. I have similar concerns for the β-catenin part too and I felt rescue experiments are necessary to join these dots.

Response:  A miRNA can target many genes.  Although we identified CCNG2, FOXO3 and FOXA2 are direct targets of miR-590-3p, it is highly possible that other genes are also involved in the tumor-promoting effects of miR-590-3p.  We have previously reported that FOXO3 is a transcriptional inducer of CCNG2 (Fu and Peng, Oncogene, 2011).  We, therefore, suggest that FOXO3 exerts its effect via CCNG2.  However, FOXO3 has been reported to have tumor-suppressive effects by regulating many genes in various types of cancers, including ovarian cancer.  It is possible that FOXO3 exerts anti-tumor effects via CCNG2 and other target genes. We have modified the Discussion to explain the possible mechanisms by which FOXO3 inhibits EOC development (lines 337-343).  Finally, we showed the silencing of CTNNB1 attenuated the effect of mir-590 on the formation of compact spheroids.  This supports our conclusion that up-regulation of the beta-catenin signaling is a key event in miR-590-3p-induced ovarian cancer development. 

In fig. 3: Its difficult to understand why these is a difference in the assay readouts in these siRNA of miRNA transfection conditions. I think the trends should look similar in all the assays.

Response: Although transient transfection of siCCNG2, siFOXO3 and miR-590-3p all increased cell proliferation, migration, and invasion, their relative effectiveness varied in these experiments.  This could be due to the different cell lines used, different transfection efficiency, or different ECL brand and exposure time used for each experiment.  Results presented in Fig. 3B and 3C were obtained from ES-2 cells while data presented in Fig. 3D was from SKOV3.ip1 cells.  Since the siRNAs and miR-590-3p were transfected into the cells transiently, their transfection and/or knockdown efficiencies may vary among different experiments.  The key point here is that silence of CCNG2 and FOXO3 or force overexpression of miR-590-3p all increased proliferation, migration, and invasion.

Minor Comments;

Line 31: EOC full form missing.

Response: EOC is now defined in line 31.

Line 93: Sentence missing.

Response: fixed.

In all the figure legends, which statistical methods have been used are missing.

Response: added.

Line 112: Information regarding siRNA transfection conditions and timings missing in the methods section.

Response: we have added the information (lines 430-432).

Fig. 3A: The right top panel of the blot is of FOXO3?

Response: Yes, it is now labeled.

Significance values are missing in many of the graphs.

Response: We have changed the way statistical significances are denoted.  The values are indicated by the number of * now.

Reviewer 4 Report

Dear Authors,

Congratulations on the successful completion and presentation of results of the study titled "miR-590-3p Targets Cyclin G2 and FOXO3 to Promote Ovarian Cancer Cell Proliferation, Invasion, and Spheroid Formation" which investigated the role of miR-590-3p in promoting ovarian cancer growth. The manuscript is well structured and the results have been presented in an easy to understand and follow format. However, the manuscript lacks some important supporting data, methodological issues and grammatical errors which need to addressed and rectified in order to make the manuscript suitable for publication in Cancers

Introduction: The authors have to expand abbreviations such as EOC (mentioned in second line of introduction) before using them in the text. 

Figure 1: The authors could briefly explain the logical reason behind the usage of mir-590, a precursor of miR-590-3p, in the results section, to help non-expert audience understand the experiments involving mir-590 and the expected outcomes

 Figure 2: The level of FOXO3 protein (band) in the western blots across different experiments seem to vary considerably. The authors have to mention whether the initial amount of lysate used in each experiment was similar or different, and if yes, the reason for the variability in FOXO3 protein level needs to be explained. 

Figure 2D: The authors could present the data for protein level of FOXO3 in siFOXO3 cells with a western blot to confirm the reduction of FOXO3 protein after siFOXO3.

Figure 3A: The authors have mentioned in the legend that the western blot after siCCNG2 and siFOXO3 confirmed decrease in CCNG2 and FOXO3 levels. But the western blot presented shows very very minimal reduction of CCNG2 (left panel) and although the right panel seems to show FOXO3 protein level under siFOXO3, but the labeling seems to be wrong which needs to corrected. Also, showing the mRNA levels of both FOXO3 and CCNG2 after siRNA knockdown is important.

 Figure 4: The authors need to show protein and/or mRNA levels of over-expressed CCNG2 and FOXO3 before showing the cell proliferation data. Also, mentioning the name of cell line used for the experiment in the figure legend will make it easy to follow the results.

Figure 5: Again, showing the protein over-expression data for CCNG2, FOXO3 and FOXO2 proteins is important here to substantiate the results. Although a schema depicting the hanging drop culture methodology is not necessary but may be helpful .

Finally, has it been reported/shown previously, that miR-590-3p is over-expressed or associated with tumor aggressiveness in human cancer patients affected by cancers other than ovarian cancer? 

Addressing the above concerns along with spell check and grammatical correctness of the manuscript is key to improving the manuscript quality and making it suitable for Cancers.

Thank you,

Harinarayanan

Author Response

Reviewer 4:

Thank you very much for your helpful comments and suggestions. Our responses are listed below:

Introduction: The authors have to expand abbreviations such as EOC (mentioned in second line of introduction) before using them in the text. 

Response: EOC is now defined in line 31.

Figure 1: The authors could briefly explain the logical reason behind the usage of mir-590, a precursor of miR-590-3p, in the results section, to help non-expert audience understand the experiments involving mir-590 and the expected outcomes

Response: We have added some information about the stable cell lines at the beginning of the Results section, lines 103-106.

Figure 2: The level of FOXO3 protein (band) in the western blots across different experiments seem to vary considerably. The authors have to mention whether the initial amount of lysate used in each experiment was similar or different, and if yes, the reason for the variability in FOXO3 protein level needs to be explained. 

Response: These were 3 separate experiments.  The difference could be due to different cell lines used and/or different exposure time of the film. The most important information is that FOXO3 level is decreased when we transfected cells with miR-590-3p mimics or its precursor mir-590.  We have now included the densitometry reading in the graphs

Figure 2D: The authors could present the data for protein level of FOXO3 in siFOXO3 cells with a western blot to confirm the reduction of FOXO3 protein after siFOXO3.

Response: The qPCR and the western blot data for the siFOXO3 used in this study are provided in Fig. 2D and Fig. 3B, respectively.

Figure 3A: The authors have mentioned in the legend that the western blot after siCCNG2 and siFOXO3 confirmed decrease in CCNG2 and FOXO3 levels. But the western blot presented shows very minimal reduction of CCNG2 (left panel) and although the right panel seems to show FOXO3 protein level under siFOXO3, but the labeling seems to be wrong which needs to corrected. Also, showing the mRNA levels of both FOXO3 and CCNG2 after siRNA knockdown is important.

Response: The label for Fig. 3A is now corrected. We also performed densitometry and include the reading on the graph. qPCR data for siFOXO3 were presented in Fig. 2D. We now included qPCR data for siCCNG2 in Fig. 3A.  These siRNAs have been used in our previous studies and their validation has been reported (references 27 and 21).

Figure 4: The authors need to show protein and/or mRNA levels of over-expressed CCNG2 and FOXO3 before showing the cell proliferation data. Also, mentioning the name of cell line used for the experiment in the figure legend will make it easy to follow the results.

Response: We have added Western blots in Fig. 4A to show overexpression. The CCNG2 antibody is no longer available from the company and therefore, we used an anti-flag antibody to detect the flag tag on the CCNG2 construct.

Figure 5: Again, showing the protein over-expression data for CCNG2, FOXO3 and FOXO2 proteins is important here to substantiate the results. Although a schema depicting the hanging drop culture methodology is not necessary but may be helpful.

Response: CCNG2 and FOXO3 overexpression data are now presented in Fig. 4A. FOXA2 overexpression is shown in Fig. 5E.  We adopted the hanging drop culture from Sodek et al (reference 7) and there was a schematic presentation of the method in that paper.     

Finally, has it been reported/shown previously, that miR-590-3p is over-expressed or associated with tumor aggressiveness in human cancer patients affected by cancers other than ovarian cancer? 

Response: We include more information about miR-590-3p in cancer in the Introduction (line 47-55).

Addressing the above concerns along with spell check and grammatical correctness of the manuscript is key to improving the manuscript quality and making it suitable for Cancers.

Response: We have checked carefully and corrected some typos and errors.

Response: We have changed the way statistical significances are denoted.  The values are indicated by the number of * now. 

Round 2

Reviewer 3 Report

The authors have addressed most of concerns raised in previous review. The manuscript can be accepted for publication in IJMS. 

Reviewer 4 Report

Dear Authors, 

Congratulations on improving the manuscript. All the questions are answered and changes are acceptable. In Fig.5E the blot could have a lower exposure. Now the paper is suitable for publication.